# Synthesis and Determination of Thermotropic Liquid Crystalline Behavior of Cinnamaldehyde-Based Molecules with Two Schiff Base Linking Units

**DOI:** 10.3390/molecules25173780

**Published:** 2020-08-20

**Authors:** Zuhair Jamain, Nor Faizawani Omar, Melati Khairuddean

**Affiliations:** 1Faculty of Science and Natural Resources, Universiti Malaysia Sabah (UMS), Kota Kinabalu 88400, Malaysia; 2School of Chemical Sciences, Universiti Sains Malaysia (USM), Penang 11800, Malaysia; fae_zawani@yahoo.com

**Keywords:** Schiff base, cinnamaldehyde, liquid crystal, nematic, mesogenic

## Abstract

A series of liquid crystal molecules with two Schiff base linking units and a cinnamaldehyde core with different terminal groups were synthesized and characterized. The intermediates of 4-heptyloxybenzaldehyde (**1a**) and 4-dodeyloxybenzaldehyde (**1b**) were synthesized through the alkylation of 4-hydroxybenzaldehyde with a series of bromoalkane. A condensation reaction of cinnamaldehyde, 1,4-phenylenediamine and a series of substituted benzaldehydes with different terminal groups such as bromo, chloro, hydroxy, cinnamaldehyde, hydrogen, methoxy, heptyloxy and dodecyloxy produced a series of new cinnamaldehyde-based compounds, **2**–**9**, respectively. All these compounds were characterized using Fourier transform infrared (FTIR) spectroscopy, nuclear magnetic resonance (NMR) spectroscopy, and CHN elemental analysis. The liquid crystal properties of these compounds were determined using polarized optical microscopy (POM), and their transitions were further confirmed using differential scanning calorimetry (DSC). Compounds with chloro, bromo, methoxy, heptyloxy, and dodecyloxy substituents are mesogenic compounds with nematic phase behavior. However, the other compounds were found to be non-mesogenic without any mesophase transitions. The structure–property relationship was investigated in order to study the effect of different terminal groups and Schiff base linking units on the liquid crystalline behavior of these compounds.

## 1. Introduction

Nowadays, materials forming liquid crystal (LC) phases have found wide application in the manufacturing of displays [1], spatial light modulators [2], optical connectors [3], switches [4], and molecular sensors and detectors [5,6]. The strong demand for new LC applications has led to the synthesis of a wide range of molecules—in particular, thermotropic LC molecules [7,8]. Thermotropic LC is temperature dependent. Thermotropic LC can be characterized by the various phase transitions that occur during heating, displaying temperature transitions between the crystal and the liquid (isotropic) phases. The transition temperature from the crystal to the liquid crystal phase is known as the melting temperature (T_m_), while the temperature when the liquid crystal changes to an isotropic liquid is known as the clearing temperature (T_c_) [9]. Mesophases for thermotropic LC molecules occur at a certain temperature range. The thermal motion destroys the delicate ordering of the liquid crystal phase at high temperatures and causes the material to change into the isotropic phase.

Most thermotropic liquid crystals are calamitic molecules which have a rigid core with two or more phenyl rings and one or more flexible terminal alkyl chains. A calamitic molecule must be geometrically anisotropic, adopting a rod-shaped structure [10]. The shape and the resulting forces of the molecule give rise to the formation of the liquid crystal phases. A calamitic liquid crystal usually exhibits the nematic and smectic phases, while the discotic liquid crystal tends to form columnar and nematic phases. The nematic phase is the least-ordered mesophase and thus resembles the isotropic liquid state. The molecules have no positional order but possess an orientational order. On the contrary, molecules in the smectic phase possess both positional and orientational orders due to their arrangement in layers. Molecules in the smectic A phase have the director parallel to the layer normal, whereas molecules in the smectic C phase are tilted with respect to the layer normal [11]. The smectic A phase has a focal-conic fan texture, while the smectic C phase exhibits a broken focal-conic fan texture which can be observed under POM. A typical calamitic liquid crystal molecule has an idealized molecular structure, as shown in Figure 1.

The linking unit between ring systems increases the length of the molecules, as well as altering the polarizability and flexibility of the molecules. Linking units can impart polarity or act as non-polar groups and, hence, increase or decrease polarizability. The Schiff base is one of the most well-known linking groups used in connecting the rigid core groups. The Schiff base contains a carbon–nitrogen double bond connected to an alkyl or aryl group but not the hydrogen. Although it provides a stepped core structure, it maintains molecular linearity, and therefore provides better stability to induce the formation of the mesophase [12,13,14]. Since then, the Schiff base liquid crystals have been studied and the area has been explored extensively [15,16,17]. A variety of molecules with Schiff base units have been synthesized and their liquid crystal properties were determined [18,19].

Cinnamaldehyde is an interesting core used in the synthesis of liquid crystal molecules. Together with a Schiff base unit, the molecule provides a conjugated system. However, not many works on such conjugation have been documented. Cinnamic acid derivatives are known as mesogenic substances. The 4-alkoxycinnamic acid compounds were among the first to be reported (Figure 2). Derivatives with alkoxy chains of *n* = 1 to 16 were reported to exhibit liquid crystal properties in which the shorter chains (*n* < 9) showed nematic phase behavior while the longer chains displayed (*n* > 9) both smectic and nematic phase behavior. Interestingly, only the *E* isomers showed liquid crystal phase behavior while *Z* isomers are non-mesogenic [20].

The synthesis of cinnamylidene-*p*-octyloxyaniline was reported but it was found to be non-mesogenic. However, terminal nitro groups help to induce liquid crystal properties. Compounds **A**–**C** (Figure 3) with long flexible chains showed smectic phase behavior. Hydrogen bonding in compound **D** was investigated and higher enthalpy and melting temperatures were reported. Compound **E** possessed higher melting and clearing temperatures than compound **A**. Another interesting observation is the effect of the central spacer length in the molecules on the transition temperature. It was reported that longer tetraethylene glycol spacers (compound **H**) possessed lower melting and clearing temperatures compared with those in compounds **G** and **F**. This also lowered the thermal transition temperatures [21]. No compounds similar to these have been reported on the synthesis and properties of liquid crystals.

Previous studies focused on the synthesis of a series of thermotropic rod-like liquid crystal molecules with a cinnamaldehyde and aromatic cores attached by two Schiff base linking units. To date, there has been much research on thermotropic liquid crystalline compounds. This research focused on the optical measurements, ion-responsive properties, photoconductivity, and dielectric studies [22,23,24,25]. However, there are no studies on cinnamaldehyde-based compounds bearing different terminal groups at the periphery. The rigidity of the cinnamaldehyde core system is able to increase the thermal behavior of the molecules. As the thermal properties increase, this molecule can act as a fire retardant which can withstand high temperature ranges [26]. The effects of terminal groups on liquid crystalline behavior have also been investigated. Terminal groups that extend the molecular long axis without increasing the molecular width increase the thermal stability of the nematic phase. The nature of the terminal substituents or end groups in the mesogenic molecule has a profound influence on the liquid crystal properties of the compound. Moreover, the synthesized molecules with conjugated Schiff base units attached to a cinnamaldehyde core system have not been reported. The main interest of this study is to understand the effect of the skeleton structure of these types of molecules and their ability to induce the liquid crystal mesophase.

## 2. Results and Discussion

### 2.1. Synthesis of Intermediates and Cinnamaldehyde Compounds

The studies involved the alkylation reaction (Scheme 1) of *p*-hydroxybenzaldehyde with alkyl bromide in the presence of potassium carbonate and potassium iodide to produce intermediates **1a**–**b**. The reaction of 1,4-phenylenediamine and cinnamaldehyde with a series of benzaldehydes produced eight Schiff base compounds (**2**–**9**) with different substituents (Scheme 2). The structures of these intermediates and compounds were successfully characterized using Fourier transform infrared (FTIR) spectroscopy, nuclear magnetic resonance (NMR) spectroscopy, and CHN elemental analysis. The melting point and yield of compounds **2**–**9** are illustrated in Table 1. Meanwhile, as summary of the data on these intermediates and compounds is presented in Section 3.3.

### 2.2. FTIR Spectral Discussion

Intermediates **1a**–**b** showed similar absorption bands in the IR data. The IR data displayed absorption bands at approximately 2860 and 2922 cm^−1^ which are assigned for the symmetrical and asymmetrical C-H (sp^3^) stretching, attributed to the methylene (CH_2_) and methyl (CH_3_) groups of the heptyl and dodecyl chains. A small shoulder at 2701 cm^−1^ is the C-H stretching of the aldehyde (H-C=O). Absorptions at 1691, 1600 and 1157 cm^−1^ are assigned for the C=O, C=C and C-O stretching, respectively. No broad absorption at 3200–3300 cm^−1^ for O-H stretching is observed, which confirmed that the *p*-hydroxybenzaldehyde used as starting material in the alkylation reaction has been completely utilized. FTIR data on *p*-hydroxybenzaldehyde, **1a**, and **1b** are provided in the Appendix A section.

A further reaction of cinnamaldehyde, 1,4-phenylenediamine and a series of *p*-substituted benzaldehydes produced compounds **2–9**, as shown in Scheme 2. Compounds **2**–**7** showed similar absorption bands in the IR data. The absorptions in the region 3100 cm^−1^ refer to the C-H (sp^2^) stretching, which is attributed to the C=C-H in the aromatic ring of the phenyl core, while the aromatic C=C stretching is at 1610 cm^−1^. The absorption for C=N stretching is at 1630, indicating that the formation of Schiff base linking units was a success. Compound **7** showed symmetrical and asymmetrical C-H (sp^3^) stretching at 2878 and 2930 cm^−1^, whereas the band at 1100 cm^−1^ was assigned for C-O stretching. Meanwhile, compounds **8** and **9** showed an additional strong signal for the heptyl and dodecyl terminal chains, respectively. The IR spectrum of compound **8** will be used as an example (Figure 4). The IR spectrum showed an absorption band at 3080 cm^−1^, which is assigned for the C-H (sp^2^) stretching, attributed to the aromatic and ethylene protons, while the bands at 2880 and 2936 cm^−1^ are assigned for the symmetrical and asymmetrical C-H (sp^3^) stretching, attributed to the methylene (CH_2_) and methyl (CH_3_) groups of the heptyl chains. A band at 1637 cm^−1^ assigned for C=N stretching confirmed the formation of the Schiff base units in compound **8**. Meanwhile, the bands at 1617, 1251, and 1140 cm^−1^ were assigned to C=C, C-N, and C-O stretching, respectively. The overall FTIR data on compounds **2**–**9** are summarized in Table 2.

### 2.3. NMR Spectral Discussion

Compound **8** was used to represent the structure confirmation in this series. The structure of compound **8** with complete atomic numbering is shown in Figure 5.

The ^1^H-NMR spectrum of compound **8** (Figure 6) showed the presence of two azomethine protons, six doublets for aromatic protons and a methyl proton in the upfield region. The presence of 12 methylene in compound **8** is attributable to the heptyl chains. H7 and H8 appeared as one signal at 7.28 ppm since they experience a similar electron environment. The doublet of doublet signal at 7.13 ppm is assigned to H5 because this proton has the potential to cause double splitting.

The ^13^C-NMR spectrum (Figure 7) of compound **8** showed the presence of 23 carbons—one methyl, six methylene, nine aromatic, five quaternary and two azomethine carbons. The DEPT experiment was then conducted in order to confirm each type of carbon.

The DEPT-90 spectrum (Figure 8a) can distinguish the methyl carbon signal at δ 13.1 ppm since this signal disappeared in the spectrum but it appeared in the DEPT-135 spectrum (Figure 8b). The negative signals in DEPT-135 are assigned for six methylene carbons in the heptyl chain. The signals assigned as quaternary carbons at δ 161.3, 148.9, and 135.1 ppm did not appear in the DEPT spectra. The quaternary carbon attached to the neighboring alkoxy chain is assigned as C19 at δ 161.3 ppm due to the deshielding effect of the electronegative oxygen.

Further confirmation of the protons and their correlations with the respected carbons was achieved using 2D NMR of COSY (^1^H-^1^H) and HMQC (^1^H-^13^C) spectra, as shown in Figure 9 and Figure 10, respectively. The COSY (^1^H-^1^H) spectrum (Figure 9) revealed the connectivity between H1-H2, H4-H5-H6, H11-H10, and H12-H13. Further confirmations of ^1^H-^1^H connectivity were also observed between the methylene protons of the heptyl chain in the upfield region. Proton H9 did not give cross peaks, indicating the presence of azomethine proton. The HMQC (^1^H-^13^C) spectrum in Figure 10 confirmed the connectivity and correlations between protons and its carbon. These correlations are summarized in Table 3.

For other homologues, compounds **3**–**9** showed a slight difference in the chemical shift values of each compound due to the different chemical environment, electronegativity effect, and bond angles [31]. Only compounds **7** and **9** showed an additional signal for the methoxy and dodecyloxy chains in the ^1^H and ^13^C NMR spectra, respectively. The overall chemical shift of compounds **2**–**9** are summarized in Table 4.

### 2.4. Determination of the Liquid Crystal Mesophase

The liquid crystal properties for compounds **2**–**9** were determined using polarized optical microscopy (POM). POM displays the texture of liquid crystal properties under polarized light with controlled temperature. In this work, compounds **2**, **3**, **7**, **8**, and **9** displayed liquid crystal textures, making them as mesogenic molecules. However, compounds **4**, **5**, and **6** displayed transition textures from the crystal phase to the isotropic phase. These compounds are non-mesogenic. All the mesogenic compounds exhibited a thread-like schlieren texture with four brushes of nematic phase. Compounds **2** and **3** with halogen substituents at the terminal end showed higher mesophase transition compared to compounds **7–9** with alkoxy terminal groups. This behavior was attributed due to the high polarity and electronegativity of halogen groups which increase the lattice and melting temperatures. The liquid crystal textures observed in POM photomicrographs for mesogenic compounds **2**, **3**, **7**, **8**, and **9** are illustrated in Figure 11. Compounds **4**–**6** with hydroxy, cinnamaldehyde, and hydrogen moieties only displayed the crystal to isotopic transition and isotropic to crystal transition in both the heating and cooling cycles, respectively.

### 2.5. Determination of Thermal Transition

Further confirmation of the phase transition temperatures of the mesogenic compounds **2**, **3**, **7**, **8**, and **9** was achieved using differential scanning calorimetry (DSC). DSC thermograms were recorded in the heating and cooling cycles. The sample is heated with a scan rate of 10.0 °C/min and held at its isotropic temperature for 2 min so as to attain the thermal stability. The cooling run is performed with the same scan rate of 10.0 °C/min. The thermal enthalpy, *ΔH* (*kJ/mol*), of each phase transition was calculated. All the data are summarized in Table 5.

The DSC thermogram of compounds **2**, **3**, **7**, **8**, and **9** confirmed the existence of the nematic phase. Two endotherms were observed in both the heating and cooling cycles which referenced one type of liquid crystal phase. The first endotherm was attributed to the crystal to nematic phase transition, while the other endotherm was attributed to the nematic phase before reaching the clearing point during the heating cycle. The same trend was observed during the cooling cycle for the transitions from the I→N and the N→Cr phases. The clearing temperatures for compounds **2**, **3**, **7**, **8**, and **9** were observed at 272.28, 282.78, 222.58, 209.63, and 200.54 °C, respectively. DSC data on compounds **2** and **3** showed a wider nematic mesomorphic range as compared to compounds **7**, **8**, and **9** with alkoxy terminal chains. The behavior was attributed to the high polarity of halogen groups which are able to reduce the degree of molecular order and steric hindrance in the cinnamaldehyde central core [29]. Meanwhile, the nematic mesophase range decreased as the number of alkoxy chains increased. This phenomenon was observed in compound **7** with a wider thermal temperature range due to the less cohesive forces between the side arms and the core of the molecules [15,17]. The same trend occurred in compounds **8** and **9**. In addition, longer terminal chains lower the melting and clearing temperatures [31]. The DSC thermogram of compound **8** is illustrated in Figure 12 as an example in this series, while the DSC thermograms for other mesogenic compounds (**2**, **3**, **7**, and **9**) are provided in the Appendix A section.

### 2.6. Structure–Property Relationship

In general, a rod-like molecule is composed of flat, rigid cores, linking units, and terminal groups of long flexible chains [32]. Molecular shape is important in the self-assembly of a molecule and impacts the ordering abilities of mesogenic molecules. It is important to choose a suitable core, linking unit or terminal group to obtain a mesogenic molecule. It was reported that molecules without distinct hydrophilic and hydrophobic regions are unlikely to self-assemble into the liquid crystal phase. The π–π interactions also influence the ability of the molecules to self-assemble [33]. The synthesis of calamitic molecules with a liquid crystal mesophase requires at least two aromatic rings—either cycloaliphatic or a combination of one aromatic and one cycloaliphatic ring—which are connected directly or through a suitable linking unit. Calamitic molecules containing a rigid core have been regarded as the most suitable geometry to exhibit mesogenic behavior [34]. Therefore, the study of the structure–property relationship is very important in designing new liquid crystal materials.

The Schiff base is one of the most well-known linking units used in connecting the rigid core groups. Schiff base molecules provide a stepped core structure which can maintain molecular linearity to provide high stability and enable mesophase formation [29,31]. Liquid crystal molecules with Schiff base linking units, with two cinnamaldehyde cores attached, did not show any liquid crystal properties. However, replacing one of the cinnamaldehyde cores with a *p*-substituted benzene ring helps to induce the liquid crystal properties. It was found that substituents such as the halogen (**2**,**3**) and alkoxy groups (**7**–**9**) displayed liquid crystal properties of the nematic phase. Halogen groups are polar substituents possessing strong dipole moments, and thus have the ability to promote mesomorphic properties. The increased dipole moment can enhance the stability of lattice and melting temperatures [34]. Moreover, the electronegativity of the halogen atom reduces the degree of molecular order, which enhances a higher clearing temperature [31,35]. As a result, compounds **2** and **3** showed nematic phase behavior at high temperatures. In terms of the alkoxy substituents, the long alkyl chains add flexibility to the rigid core and stabilize the molecular interactions needed for the formation of a liquid crystal mesophase. This in turn, reduces the melting temperature, T_m_ [29]. Compound **7** with a methoxy group displayed nematic phase behavior as the molecules tend to orientate in parallel arrangement with an increase in the ionic radius of the terminal substituents [35]. Further, the length of the alkoxy side chain showed a strong influence on mesophase formation [36,37,38]. Compounds **8** and **9** did not show smectic phase transition even though these compounds attached to long terminal alkyl chains. This behavior was due to the high longitudinal polarizability and conjugation between the terminal chains and the aromatic system in cinnamaldehyde structure. Hence, only the nematic phase was facilitated in compounds **8** and **9**.

In this research, the behavior of compounds **4**–**6** attached to small terminal substituents such as hydroxy, cinnamaldehyde, and hydrogen groups did not show any liquid crystal phases. Terminal substituents can both attract and repel one another in different molecules. They can also affect the polarizability of the aromatic rings to which they are attached [39]. Hydrogen and cinnamaldehyde groups are non-polar compounds, which decreased the molecular aromaticity and polarizability of the molecules. As a result, the mesophase transition cannot be induced. Meanwhile, the lone pair of the hydroxy groups caused the cancellation of dipole moments of the molecules, which reduced the polarizability and molecular interactions needed for the formation of the mesophases [31,40]. The POM observation of compounds **2**–**9** is summarized in Table 6.

## 3. Materials and Methods

### 3.1. Chemicals

The chemicals and solvents used in this study are cinnamaldehyde, 1,4-phenylenediamine, 4-chlorobenzaldehyde, benzaldehyde, 4-bromobenzaldehyde, 4-hydroxybenzaldehyde, 4-methoxybenzaldehyde, dimethylformamide, 1-bromoheptane, 1-bromononene, 1-bromododecane, potassium carbonate, potassium iodide, methanol, ethyl acetate, and anhydrous sodium sulphate. These chemicals and solvents were purchased from Merck, Sigma-Aldrich, Fluka, System, and Qrëc (Asia). All chemicals were used as received without further purifications.

### 3.2. Instruments

In this research, thin layer chromatography (TLC) is a technique used to monitor the reaction progress and the purity of the products. TLC identifies a compound in a mixture using a thin stationary phase supported by an inert backing. The eluent used was 20:80 of ethyl acetate:hexane. Meanwhile, FTIR spectroscopy (PerkinElmer, Waltham, MA, USA) is a technique used to determine the functional groups in a sample. Samples are scanned at a range of 700 to 4000 cm^−1^. The structure elucidation of all compounds was determined using NMR spectroscopy (Bruker, Coventry, UK), and the purity of compounds was confirmed using CHN elemental analysis (PerkinElmer, Waltham, MA, USA). Moreover, the liquid crystalline behavior of these compounds was determined using POM (Linkam, London, UK). The sample was tested using an Olympus system model bx53 linksys32 with a scan rate of 10 °C and held at 2 °C as the sample reached the liquid crystal state. Finally, the mesophase transition of these compounds was confirmed using DSC (PerkinElmer, Waltham, MA, USA).

### 3.3. Synthesis Method

#### 3.3.1. Synthesis of 4-heptyloxybenzaldehyde, **1a**

The 4-Hydroxybenzaldehyde (0.10 mol) and 1-bromoheptane (0.10 mol) were dissolved in 20 mL of DMF each and both solutions were mixed in a 250 mL round-bottomed flask. Potassium carbonate (K_2_CO_3_) (0.15 mol) and potassium iodide (KI) (0.01 mol) were added into the mixture. The mixture was refluxed at 80 °C for 12 h and the reaction progress was monitored by TLC. Upon completion, the reaction mixture was poured into 250 mL of cold water and was extracted by 30 mL ethyl acetate (2×). These combined organic layers were dried in anhydrous sodium sulphate. The crude extract was filtered and evaporated to give a yellowish oil product. The same method was used to synthesize **1b**. Yield: 15.1 g (72.7 %), light-yellow oil. IR (cm^−1^) 2928 and 2734 (C*_sp3_*-H stretch), 2700 (H-CO, aldehydic), 1687 (C=O stretch), 1599 (C=C stretch), and 1157 (C-O stretch). ^1^H-NMR (500 MHz, DMSO-d_6_) δ, ppm: 9.85 (s, 1H), 7.82 (d, *J* = 8.5 Hz, 2H), 7.03 (d, *J*8.5 Hz, 2H), 3.98 (t, *J*6.5 Hz, 2H), 1.64–1.70 (m, 2H), 1.31–1.37 (m, 2H), 1.20–1.27 (m, 6H), and 0.81 (t, *J*7.0 Hz, 3H). ^13^C-NMR (125 MHz, DMSO-d_6_) δ, ppm: 190.6, 163.6, 131.4, 129.5, 114.6, 67.9, 31.2, 28.5, 28.5, 25.4, 22.1, and 13.7.

The 4-Dodecyloxybenzaldehyde, **1b;** Yield: 20.0 g (72.1%), dark yellow oil. FTIR (cm^−1^) 2922 and 2733 (C*_sp3_*-H stretch), 2701 (H-CO, aldehydic), 1691 (C=O stretch), 1600 (C=C stretch), and 1157 (C-O stretch). ^1^H-NMR (500 MHz, DMSO-*d*_6_) δ, ppm: 9.81 (s, 1H), 7.76 (d, *J* = 8.5 Hz, 2H), 6.96 (d, *J* = 8.5 Hz, 2H), 3.93 (t, *J* = 6.5 Hz, 2H), 1.61–1.67 (m, 2H), 1.31–1.35 (m, 2H), 1.15–1.22 (m, 16H), and 0.78 (t, *J* = 7.0 Hz, 3H). ^13^C-NMR (125 MHz, DMSO-*d*_6_) δ, ppm: 191.4, 163.5, 131.4, 129.5, 114.4, 67.8, 31.4, 29.1, 29.2, 29.2, 29.1, 28.9, 28.9, 28.6, 25.5, 22.1, and 13.6.

#### 3.3.2. Synthesis of *N*-(4-bromobenzylidene)-*N*′-(3-phenylallylidene)-benzen-1,4-diamine, **2**

A mixture of cinnamaldehyde (0.004 mol), 1,4-phenylenediamine (0.004 mol), and 4-bromobenzaldehyde (0.004 mol) was stirred in 15 mL methanol at room temperature. The reaction was monitored using TLC. After 5 h, the reaction was stopped, filtered, and washed with cold methanol. The crude extract was dried and recrystallization from methanol gave a yellowish powder. The same method was used to synthesize compounds **3**–**9**. Yield: 1.4 g (89.1%), mp: 208.2–210.4 °C, light yellow powder. FTIR (cm^−1^): 3010 (C*_sp2_*-H stretch), 1637 (C=N stretch), 1618 (C=C stretch), 1251 (C-N stretch). ^1^H-NMR (500 MHz, DMSO-*d*_6_) δ, ppm: 8.67 (s, 1H), 8.47 (d, *J* = 9.0 Hz, 1H), 7.89 (d, *J* = 9.0 Hz, 2H), 7.74 (d, *J* = 7.0 Hz, 2H), 7.68 (d, *J* = 7.5 Hz, 2H), 7.44 (t, *J* = 7.0 Hz, 2H), 7.36–7.39 (m, 4H), 7.28 (t, *J* = 4.5 Hz, 2H), and 7.18 (d, *J* = 5.0 Hz, 1H). ^13^C-NMR (125 MHz, DMSO-*d*_6_) δ, ppm: 161.4, 158.8, 149.5, 148.7, 144.1, 135.4, 135.2, 131.8, 130.3, 128.9, 128.4, 127.5, 124.8, 122.0, and 121.8. CHN elemental analysis: calculated for C_22_H_17_N_2_Br: C: 67.88%, H: 4.37%, and N: 7.20%. Found: C: 67.79%, H: 4.31%, and N: 7.16%.

*N*-(4-chlorobenzylidene)-*N′*-(3-phenylallylidene)-benzene-1,4-diamine, **3**; Yield: 1.3 g (87.8%), mp: 211.1–213.4 °C, yellow powder. FTIR (cm^−1^): 3050 (C*_sp2_*-H stretch), 1640 (C=N stretch), 1618 (C=C stretch), and 1251 (C-N stretch). ^1^H-NMR (500 MHz, DMSO-*d*_6_) δ, ppm: 8.70 (s, 1H), 8.47 (d, *J* = 8.5 Hz, 1H), 7.97 (d, *J* = 6.0 Hz, 2H), 7.67 (d, *J* = 7.0 Hz, 2H), 7.60 (d, *J* = 7.5 Hz, 2H), 7.44 (t, *J* = 7.0 Hz, 2H), 7.36–7.39 (m, 4H), 7.29–7.35 (m, 2H), and 7.18 (dd, *J* = 8.5, 12.0 Hz, 1H). ^13^C-NMR (125 MHz, DMSO-*d*_6_) δ, ppm: 161.4, 158.6, 149.5, 148.7, 144.1, 135.4, 135.2, 131.8, 130.3, 128.9, 128.4, 127.5, 124.8, 122.0, and 121.8. CHN elemental analysis: calculated for C_22_H_17_N_2_Cl: C: 76.63%, H: 4.93%, and N: 8.15%. Found: C: 76.55%, H: 4.82%, and N: 8.13%.*N*-(4-hydroxybenzylidene)-*N′*-(3-phenylallylidene)-benzene-1,4-diamine, **4**; Yield: 1.1 g (81.5%), mp: 196.8–199.1 °C, yellow powder. FTIR (cm^−1^): 3414 (O-H stretch), 3060 (C*_sp2_*-H stretch), 1634 (C=N stretch), 1618 (C=C stretch), and 1253 (C-N stretch). ^1^H-NMR (500 MHz, DMSO-*d*_6_) δ, ppm: 8.51 (s, 1H), 8.45 (d, *J* = 9.0 Hz, 1H), 7.77 (d, *J* = 9.0 Hz, 2H), 7.66 (d, *J* = 7.0 Hz, 2H), 7.44 (t, *J* = 6.0 Hz, 2H), 7.39 (t, *J* = 7.5 Hz, 2H), 7.25 (s, 4H), 7.13 (dd, *J* = 8.5, 16.0 Hz, 2H), and 6.88 (d, *J* = 6.5 Hz, 1H). ^13^C-NMR (125 MHz, DMSO-*d*_6_) δ, ppm: 162.1, 161.1, 160.5, 149.4, 143.9, 135.6, 130.5, 129.5, 128.9, 128.6, 127.5, 121.8, 121.7, and 115.7. CHN elemental analysis: calculated for C_22_H_18_N_2_O: C: 80.98%, H: 5.52%, and N: 8.59%. Found: C: 80.91%, H: 5.49%, and N: 8.57%.*N*-(3-phenylallylidene)-*N′*-(3-phenylallylidene)-benzene-1,4-diamine, **5**; Yield: 1.0 g (76.9%), mp: 190.5–193.3 °C, yellow powder. FTIR (cm^−1^): 3075 (C*_sp2_*-H stretch), 1635 (C=N stretch), 1600 (C=C stretch), and 1240 (C-N stretch). ^1^H-NMR (500 MHz, DMSO-*d*_6_) δ, ppm: 8.45 (d, *J* = 8.5 Hz, 1H), 7.66 (d, *J* = 7.5 Hz, 2H), 7.35–7.39 (m, 2H), 7.44 (t, *J* = 6.0 Hz, 2H), 7.25 (s, 4H), and 7.16 (dd, *J* = 9.0, 16.0 Hz, 1H). ^13^C-NMR (125 MHz, DMSO-*d*_6_) δ, ppm: 160.6, 149.8, 143.3, 135.2, 129.0, 128.5, 128.2, 127.0, and 121.3. CHN elemental analysis: calculated for C_24_H_20_N_2_: C: 85.71%, H: 5.81%, and N: 9.03%. Found: C: 85.69%, H: 5.84%, and N: 8.97%.*N*-benzylidene-*N′*-(3-phenylallylidene)-benzene-1,4-diamine, **6**; Yield: 0.9 g (72.6%), mp: 201.5–203.3 °C, light brown powder. FTIR (cm^−1^): 3060 (C*_sp2_*-H stretch), 1637 (C=N stretch), 1617 (C=C stretch), and 1248 (C-N stretch). ^1^H-NMR (500 MHz, DMSO-*d*_6_) δ, ppm: 8.70, (s, 1H), 8.47 (d, *J* = 8.5 Hz, 1H), 7.96 (d, *J* = 7.0 Hz, 2H), 7.66 (d, *J* = 6.5 Hz, 2H), 7.53 (d, *J* = 1.5 Hz, 2H), 7.52 (t, *J* = 1.5 Hz, 1H), 7.44 (t, *J* = 8.5 Hz, 2H), 7.35–7.38 (m, 4H), 7.25 (s, 2H), and 7.14 (dd, *J* = 8.5, 15.0 Hz, 1H). ^13^C-NMR (125 MHz, DMSO-*d*_6_) δ, ppm: 161.5, 160.3, 150.0, 149.9, 144.2, 136.8, 136.2, 131.7, 129.9, 129.4, 129.2, 129.1, 129.0, 128.0, 122.4, and 122.2. CHN elemental analysis: calculated for C_22_H_18_N_2_: C: 85.16%, H: 5.81%, and N: 9.03%. Found: C: 85.09%, H: 5.79%, and N: 8.99%.*N*-(4-methoxybenzylidene)-*N′*-(3-phenylallylidene)-benzene-1,4-diamine, **7**; Yield: 1.0 g (74.3%), 174.8–177.5 °C, yellow powder. FTIR (cm^−1^): 3070 (C*_sp2_*-H stretch), 2930 and 2878 (C*_sp3_*-H stretch), 1640 (C=N stretch), 1618 (C=C stretch), 1251 (C-N stretch), and 1100 (C-O stretch). ^1^H-NMR (500 MHz, DMSO-*d*_6_) δ, ppm: 8.59 (s, 1H), 8.47 (d, *J* = 9.0 Hz, 1H), 7.89 (d, *J* = 11.0 Hz, 2H), 7.68 (d, *J* = 8 Hz, 2H), 7.46 (d, *J* = 6.3 Hz, 2H), 7.35–7.41 (m, 2H), 7.27–7.30 (m, 4H), 7.18 (dd, *J* = 8.7, 15.0 Hz, 2H), 7.06 (d, *J* = 9.0 Hz, 1H), and 3.84 (s, 3H). ^13^C-NMR (125 MHz, DMSO-*d*_6_) δ, ppm: 162.2, 161.6, 159.9, 148.4, 143.1, 135.3, 131.3, 129.8, 128.4, 122.7, and 115.1. CHN elemental analysis: calculated for C_23_H_20_N_2_O: C: 81.18%, H: 5.88%, and N: 8.23%. Found: C: 81.08%, H: 5.82%, and N: 8.18%.*N*-(4-heptyloxybenzylidene)-*N′*-(3-phenylallylidene)-benzene-1,4-diamine, **8**; Yield: 1.3 g (74.9%), mp: 166.4–169.8 °C, yellow powder. FTIR (cm^−1^): 3080 (C*_sp2_*-H stretch), 2936 and 2880 (C*_sp3_*-H stretch), 1637 (C=N stretch), 1617 (C=C stretch), 1251 (C-N stretch), and 1140 (C-O stretch). ^1^H-NMR (500 MHz, DMSO-*d*_6_) δ, ppm: 8.57 (s, 1H), 8.45 (d, *J* = 9 Hz, 1H), 7.87 (d, *J* = 7.5 Hz, 2H), 7.69 (d, *J* = 7.5 Hz, 2H), 7.44 (t, *J* = 7.5 Hz, 2H), 7.35–7.38 (m, 2H), 7.27 (s, 4H), 7.13 (dd, *J* = 9, 15.0 Hz, 1H), 7.05 (d, *J* = 8.5 Hz, 2H), 4.08 (t, *J* = 6.5 Hz, 2H), 1.75 (t, *J* = 7Hz, 2H), 1.44–1.46 (m, 2H), 1.31–1.37 (m, 6H), and 0.89 (t, *J* = 6 Hz, 3H). ^13^C-NMR (125 MHz, DMSO-*d*_6_) δ, ppm: 161.1, 160.4, 158.2, 148.9, 143.1, 135.1, 129.8, 128.8, 128.6, 128.3, 128.1, 126.9, 121.2, 114.3, 67.4, 30.6, 28.1, 27.7, 24.8, 21.3, and 13.1. CHN elemental analysis: calculated for C_29_H_32_N_2_O: C: 82.08%, H: 7.55%, and N: 6.60%. Found: C: 82.06%, H: 7.48%, and N: 6.56%.*N*-(4-dodecyloxybenzylidene)-*N′*-(3-phenylallylidene)-benzene-1,4-diamine, **9**; Yield: 1.5 g (76.9%), mp: 162.1–165.9 °C, yellow powder. FTIR (cm^−1^): 3070 (C*_sp2_*-H stretch), 2936 and 2880 (C*_sp3_*-H stretch), 1637 (C=N stretch), 1617 (C=C stretch), 1250 (C-N stretch), and 1170 (C-O stretch). ^1^H-NMR (500 MHz, DMSO-*d*_6_) δ, ppm: 8.56 (s, 1H), 8.45 (d, *J* = 9.0 Hz, 1H), 7.78 (d, *J* = 7.5 Hz, 2H), 7.66 (d, *J* = 7.0 Hz, 2H), 7.35–7.44 (m, 4H), 7.28 (s, 4H), 7.13 (dd, *J* = 10.0, 15.0 Hz, 1H), 7.05 (d, *J* = 8.0 Hz, 2H), 4.08 (t, *J* = 6.0 Hz, 2H), 1.75 (t, *J* = 6.0 Hz, 2H), 1.43–1.45 (m, 2H) 1.27–1.42 (m, 16H), and 0.87 (t, *J* = 2.0 Hz, 3H). ^13^C-NMR (125 MHz, DMSO-*d*_6_) δ, ppm: 162.3, 160.4, 158.6, 149.6, 143.2, 135.2, 130.0, 128.5, 128.3, 127.1, 121.3, 114.5, 67.6, 30.8, 28.5, 28.2, 25.0, 21.5, and 13.3. CHN elemental analysis: calculated for C_34_H_42_N_2_O: C: 82.59%, H: 8.50%, and N: 5.67%. Found: C: 82.54%, H: 8.45%, and N: 5.63%

## 4. Conclusions

Thermotropic compounds **2**–**9** with a rod-like structure of cinnamaldehyde and phenyl cores, attached with Schiff base linking units and varieties of terminal groups, were successfully synthesized and characterized. Structural characterization was confirmed using FTIR, 1D and 2D NMR, and CHN elemental analysis. Based on the POM observation, compounds **2**, **3**, **7**, **8**, and **9** were mesogenic with nematic phase behavior. Compounds **2** and **3** with polar groups (Br and Cl) at the terminal end possessed strong dipole moments and had the ability to promote mesomorphic nematic properties. Meanwhile, compounds **7**, **8**, and **9** attached to alkoxy terminal chains (CH_3_, C_7_H_15_, and C_12_H_25_) were able to add flexibility to the rigid core and stabilize the molecular interactions needed for the formation of a liquid crystal mesophase. However, compounds **4**, **5**, and **6** are non-mesogenic due to the low polarizability and molecular interactions in the molecules. The presence of a cinnamaldehyde core system led to high rigidity and resulted in high mesophase transitions, and could therefore be used to display liquid crystalline behavior at high temperature ranges.

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
