# Peer review of "Synthesis and Determination of Thermotropic Liquid Crystalline Behavior of Cinnamaldehyde-Based Molecules with Two Schiff Base Linking Units"

_molecules, 2020, doi:10.3390/molecules25173780_

Round 1

Reviewer 1 Report

The authors present the synthesis and a brief characterization of a set of chemical compounds with liquid crystal behavior. In my opinion, some sections of this manuscript must be seriously revised. Here are my observations:

  1. The title does not justify the content. According to the title, the authors should present the “Synthesis and Thermotropic Liquid Crystalline Behavior of Cinnamaldehyde-Based Molecules with Two Schiff Base Linking Units”. The synthesis process is widely presented while the liquid crystalline behavior is briefly discussed.
  2. The Materials and Methods section must be improved. The methods should be presented here, not in Results and Dicusions.
  3. In section 2.5., Determination of Thermal Transition, the authors are discussing the properties of

 the mesogenic compounds 2, 3, 7,  8, and 9 but only present the compound 8 thermogram. Besides, the nematic temperature range is not clearly

  1. In section 2.6., the authors discuss about the properties and liquid crystalline behavior of the synthesized compounds without presenting the source of their observations such as plots or data connected to their conclusions.
  2. In Conclusions section, the phrase „The presence of the polar (Br and Cl) and alkoxy (CH3, C7H15, and

C12H25) substituents at the terminal group able to induce the liquid crystal behavior of the molecules.„ has no sense. Besides the author should justify the use of nematic liquid crystalline phases around 200 Celsius degrees temperatures.

Author Response

Response to Reviewer 1 Comments

Point 1: The title does not justify the content. According to the title, the authors should present the “Synthesis and Thermotropic Liquid Crystalline Behavior of Cinnamaldehyde-Based Molecules with Two Schiff Base Linking Units”. The synthesis process is widely presented while the liquid crystalline behavior is briefly discussed.

Response 1: New title of the manuscript: Synthesis and Determination of Thermotropic Liquid Crystalline Behavior of Cinnamaldehyde-Based Molecules with Two Schiff Base Linking Units.

Point 2: The Materials and Methods section must be improved. The methods should be presented here, not in Results and Dicusions.

Response 2: The materials and methods section are located at Section 3.3. Section 2.1 with schemes is provided (not a methodology) in order to begin the discussion of results by somehow describing the compounds used in this study. This could be made by referring to Schemes 1-2 already in the beginning of presentation of results, although the detailed presentation of synthesis takes place later in Section 3.3. Readers must see the structures that the formula numbers refer to.

Point 3: In section 2.5., Determination of Thermal Transition, the authors are discussing the properties of the mesogenic compounds 2, 3, 7,  8, and 9 but only present the compound 8 thermogram. Besides, the nematic temperature range is not clearly.

Response 3: (i) Only DSC thermogram of compound 8 is shown in the manuscript as a representative of the other compounds. The DSC compounds of 2, 3, 7, and 9 are provided in the Supplementary Materials Section (Line 237-239). (ii) The discussion on the nematic temperature range is added (Line 229-237): The DSC data of compounds 2 and 3 showed a wider nematic mesomorphic range as compared to compounds 7, 8, and 9with alkoxy terminal chains. The behaviour was attributed to the high polarity of halogen groups which able to reduces the degree of molecular order and steric hindrance in the cinnamaldehyde central core [29]. Meanwhile, the nematic mesophase range decreased as the number of alkoxy chains increased. This phenomenon was observed in compound 7with wider thermal temperature range due to the less cohesive forces between the side arms and the core of the molecules [15, 17]. The same trend was occurred in compounds 8 and 9. In addition, longer terminal chains lower the melting and clearing temperatures [31].

Point 4: In section 2.6., the authors discuss about the properties and liquid crystalline behavior of the synthesized compounds without presenting the source of their observations such as plots or data connected to their conclusions.

Response 4: The discussion in Section 2.6 have been connected to their conclusions. Line 264-265: As a result, compounds 2 and 3 showed the nematic phase at high temperature. Line 268-269: Compound 7 with methoxy group displayed the behaviour of nematic phase as the molecules tend to orientate in parallel arrangement with increase in ionic radius of the terminal substituents [35]. Line 274: Hence, only nematic phase was facilitated in compounds 8 and 9.

Point 5: In Conclusions section, the phrase „The presence of the polar (Br and Cl) and alkoxy (CH3, C7H15, and C12H25) substituents at the terminal group able to induce the liquid crystal behavior of the molecules.„ has no sense. Besides the author should justify the use of nematic liquid crystalline phases around 200 Celsius degrees temperatures.

Response 5: The conclusion have been revised. Line 409-416: Compounds 2 and 3 with polar groups (Br and Cl) at the terminal end possessed strong dipole moments and thus having the ability to promote mesomorphic nematic properties. Meanwhile, compounds 7, 8, and 9 attached to alkoxy terminal chains (CH3, C7H15, and C12H25) are able to add flexibility to the rigid core and stabilize the molecular interactions needed for the formation of liquid crystal mesophase. However, compounds 4, 5, and 6 are non-mesogenic due to the low polarizability and molecular interactions in the molecules. The presence of cinnamaldehyde core system led to high rigidity and resulted in high mesophase transitions which has a potential to be used as liquid crystal display with high temperature ranges.

Reviewer 2 Report

The authors synthesized a series of Schiff base compounds with different terminal groups and characterized their chemical structure and thermotropic liquid crystallinity.  This reviewer judges that it may be published in the journal molecules with adequate minor revisions.  The reviewer's specific comments are listed below:

#1.  Introduction:  The authors must emphasize novelty and usability of the newly synthesized cinnamaldehyde-based molecules.  What is the superiority of your components, when compared with existing liquid crystal molecules?

#2.  Page 5, Lines 123–136:  Please add an explanation for Csp3-H stretch (2930, 2878 cm−1) and C-O stretch (1100 cm−1) of compound 7 in the text.

 #3.  IR data of the intermediates 1a–b:  It would be helpful to list the IR data of the intermediates 1a–b and p-hydroxybenzaldehyde used as a starting material in Supplementary Materials: It is because the authors make an animated discussion about the FTIR spectra.  #4.  NMR spectra (Figures 6–10):  Resolution of the figures is poor and the inserted numerals are hard to read.  In addition, 13C resonance signals of C6 and C9 appear at around 160 ppm in Figure 7, but in Figure 8 the signals seem to shift toward ~155 ppm.  Is it OK??

#5.  Instruments:  Please describe the measurement conditions (thermal program, heating rate, etc.) of the POM observation, in detail.  Also, equipment name of the DSC apparatus is unknown.

Mistakes(?)

- Line 115:  1a–d  =>  1a–b

- Line 125:  3400 cm−1  =>  3100 cm−1

- Figure 11:  Magnification of 10×0.4  =>  Magnification of 10×4

- Table 5:  24.41 (kJ/mol)  =>  −24.41 (kJ/mol)

Author Response

Response to Reviewer 2 Comments

Point 1: #1.  Introduction:  The authors must emphasize novelty and usability of the newly synthesized cinnamaldehyde-based molecules.  What is the superiority of your components, when compared with existing liquid crystal molecules?

Response 1: The discussions have been added. Line 89-95: To date, a lot of pervious works on the thermotropic liquid crystalline compounds have been reported. This research focussed on the optical measurements, ion-responsive properties, photoconductivity, and dielectric studies [22-25]. However, there are no research works reported on the cinnamaldehyde-based compounds bearing different terminal groups at the periphery. The rigidity of the cinnamaldehyde core system able to increase the thermal behaviour of the molecules. As the thermal properties increased, this molecule can act as a fire retardant materials which can stand at high temperature ranges [26].

Point 2: #2.  Page 5, Lines 123–136:  Please add an explanation for Csp3-H stretch (2930, 2878 cm−1) and C-O stretch (1100 cm−1) of compound 7 in the text.

Response 2: The explanation have been added. Line 135-137: Compound 7 showed the symmetrical and asymmetrical C-H (sp3) stretching at 2878 and 2930 cm-1, whereas the band at 1100 cm-1 was assigned for C-O stretching.

Point 3: #3.  IR data of the intermediates 1a–b:  It would be helpful to list the IR data of the intermediates 1a–b and p-hydroxybenzaldehyde used as a starting material in Supplementary Materials: It is because the authors make an animated discussion about the FTIR spectra. 

Response 3: The IR data of the intermediates 1a–b and p-hydroxybenzaldehyde used as a starting material have been added in Supplementary Materials Section.

Point 4: #4.  NMR spectra (Figures 6–10):  Resolution of the figures is poor and the inserted numerals are hard to read.  In addition, 13C resonance signals of C6 and C9 appear at around 160 ppm in Figure 7, but in Figure 8 the signals seem to shift toward ~155 ppm.  Is it OK??

Response 4: Figures 6-10 have been revised and restructured. The signal shift toward ~155ppm which might due to the effect of the spin-spin effect of the NMR machine. However, the different is not obvious and still acceptable.

Point 5: #5.  Instruments:  Please describe the measurement conditions (thermal program, heating rate, etc.) of the POM observation, in detail.  Also, equipment name of the DSC apparatus is unknown.

Response 5: The measurements conditions of the POM have been added. Line 302-303: The sample was tested using an Olympus system model bx53 linksys32 with a scan rate of 10 °and held at 1 °as the sample reaching the liquid crystal state. The DSC equipment data also have been added in Line 303-305: Finally, the mesophase transition of these compounds was confirmed using DSC (PerkinElmer, Waltham, MA, USA).

Point & Response 6: 

Mistakes(?)

- Line 115:  1a–d  =>  1a–b (Corrected: Line 121)

- Line 125:  3400 cm−1  =>  3100 cm−1 (Corrected: Line 132)

- Figure 11:  Magnification of 10×0.4  =>  Magnification of 10×4

(The Magnification 10 x 0.4 was correct. This measurement are provided in the POM machine)

- Table 5:  24.41 (kJ/mol)  =>  −24.41 (kJ/mol) (Corrected in Table 5: Line 221)

Round 2

Reviewer 1 Report

The authors effort to improve this manuscript is obvious and, in my opinion, their work can be considered for publication. Yet another revision is necessary because there are still some minor errors.

I found some mistakes  and I believe a careful check of the authors is needed.

Lines 94-05: The phrase “The rigidity 94 of the cinnamaldehyde core system able to increase the thermal behavior of the molecules.”, has no predicate.

Line 248: Figure 12 is inserted twice